# Occupational Exposure to Carbon Nanotubes and Carbon Nanofibres: More Than a Cobweb

**DOI:** 10.3390/nano11030745

**Published:** 2021-03-16

**Authors:** Enrico Bergamaschi, Giacomo Garzaro, Georgia Wilson Jones, Martina Buglisi, Michele Caniglia, Alessandro Godono, Davide Bosio, Ivana Fenoglio, Irina Guseva Canu

**Affiliations:** 1Unit of Occupational Medicine, Department of Public Health Sciences and Pediatrics, University of Turin, Via Zuretti 29, 10126 Torino, Italy; giacomo.garzaro@unito.it (G.G.); georgia.wilson.jones@gmail.com (G.W.J.); martina.buglisi@unito.it (M.B.); michele.caniglia@unito.it (M.C.); alessandro.godono@unito.it (A.G.); davide.bosio@unito.it (D.B.); 2Department of Chemistry, University of Turin, Via P. Giuria 7, 10125 Torino, Italy; 3Center for Primary Care and Public Health (Unisanté), Department of Occupational and Environmental Health, Route de la Corniche 2, 1066 Lausanne, Switzerland; irina.guseva-canu@unisante.ch

**Keywords:** carbon nanotubes, carbon nanofibers, occupational exposure, health effects, epidemiological studies, occupational exposure limits, risk assessment

## Abstract

Carbon nanotubes (CNTs) and carbon nanofibers (CNFs) are erroneously considered as singular material entities. Instead, they should be regarded as a heterogeneous class of materials bearing different properties eliciting particular biological outcomes both in vitro and in vivo. Given the pace at which the industrial production of CNTs/CNFs is increasing, it is becoming of utmost importance to acquire comprehensive knowledge regarding their biological activity and their hazardous effects in humans. Animal studies carried out by inhalation showed that some CNTs/CNFs species can cause deleterious effects such as inflammation and lung tissue remodeling. Their physico-chemical properties, biological behavior and biopersistence make them similar to asbestos fibers. Human studies suggest some mild effects in workers handling CNTs/CNFs. However, owing to their cross-sectional design, researchers have been as yet unable to firmly demonstrate a causal relationship between such an exposure and the observed effects. Estimation of acceptable exposure levels should warrant a proper risk management. The aim of this review is to challenge the conception of CNTs/CNFs as a single, unified material entity and prompt the establishment of standardized hazard and exposure assessment methodologies able to properly feed risk assessment and management frameworks.

## 1. Introduction

Carbon nanotubes (CNTs) are cylindrically shaped carbon-based nano-objects [1]. Recently, they have been included in the EU ‘Substitute It Now’ (SIN) list as nanomaterials of Very High Concern [2]. Since their discovery in 1991, CNTs have been utilized for an array of industrial and academic purposes because of their unique material properties, ranging from stiffness and strength to thermal and electrical conductivity. For industrial applications in particular, the three properties which are most sought after are: (i) mechanical strength—they are 5 times lighter and 20–100 times stronger than steel; (ii) electrical conductivity—they are as conductive as copper, and (iii) thermal conductivity—which is similar to diamond and twice that of copper [2]. These properties are defined by CNT characteristics such as structure, size and geometry. In fact, not all CNT have the same characteristics. This has lead experts in the field to question the justness of restrictions and bans on CNTs, urging the scientific community to recognize that CNTs do not all belong to a singular material category. Rather, in order to perform valid risk assessments, they should be subdivided in different classes on the basis of their physico-chemical similarities [3].

CNTs can differ greatly in terms of size, shape, and chemical composition, both on the basis of design and as a result of contamination during production. Firstly, they can be subdivided according to the number of carbon atoms layers which make up their wall; thus, they are classified as single-walled CNTs (SWCNTs) or multi-walled CNTs (MWCNT). SWCNTs feature diameters spanning from 0.5 to a few nanometers, whereas MWCNTs consist of one or more single coaxial pipe walls, with diameters ranging from 5 to 100–200 nm and lengths up to several millimeters. Owing to the number of graphene layers and of the arrangement of hybridized *sp2* carbon hexagons, or to the presence of manufacturing defects, CNTs can have different geometries. They can be straight and thin, curved or coiled, rigid or partially flexible. Current methods of synthesis commonly generate morphologically heterogeneous SWCNTs or MWCNTs. Moreover, depending on the dispersion medium, they can exist as single entities or as aggregates/agglomerates that behave like particles rather than fibers [1]. In order to improve their electrical and/or physical properties or to vehiculate other organic substances (for example, antineoplastic drugs), CNTs surface can be functionalized with a wide variety of chemicals.

Modern CNTs are not perfect structures. On the contrary, they contain various types of structural defects generated during the growth process or during post-synthesis treatments, such as the replacement of carbon atoms with nitrogen. Commercially available CNTs may also contain amorphous carbon, fullerenes, graphitic particles and metals derived from the catalysts used in their synthesis (e.g., Fe, Co, Ni, Y), and inert materials (e.g., alumina, silica) used as catalyst supports. Impurities can reach up to 20–30% *w/w* of the product and are known to affect CNT technological performance and biological effects [4].

Although almost 30 years have passed since their discovery, CNT areas of application are still limited, mainly due to problems with well-controlled mass production. The greatest field of application is currently represented by the production of composites and electronic applications [5]. Currently, CNTs are used in plastics to make them lighter or more resistant: with a CNT content varying from 1–10%, for the production of automobile, aircraft and wind turbine components. The global market for MWCNTs has witnessed a decline in large-scale production [5] but now it is expected to grow at a CAGR of 16.70% between 2018 and 2023 [6]. Potential future applications include textiles, energy systems, electronics, aerospace, building materials, biomedicine, and medical devices. Particularly, within the biomedical realm, potential applications range from the production of artificial tissues, scaffolds for the regeneration of bone tissue and substrates for neuronal growth [1,5,6].

## 2. Hazards from CNTs

Historically, the potential toxicity of CNTs has been underestimated, particularly because their production mainly took place in small-scale research and development (R&D) laboratories. However, an article published in 1998 in Science hypothesized a connection between CNTs and asbestos, based on similarities in both structure and biological behavior, especially with regards to biopersistence [7]. Most experimental studies have considered the effects on the respiratory system, as the inhalation route is the most relevant means of exposure in occupational settings. Exposure by inhalation, instillation or aspiration in rodents is associated with human relevant endpoints, such as lung inflammation, genotoxicity, granuloma formation and lung fibrosis [8,9]. These effects were also associated with lung cancer, thus justifying the assessment of any genotoxic endpoints related to CNT exposure.

The induction of both pleural and peritoneal mesothelial neoplasms has been demonstrated by several authors, but the resulting evidence is conflicting, ranging from mesothelioma induction [9] to the absence of any carcinogenic effects [10]. Some authors have highlighted that MWCNT >10 μm in length harbor the capacity to induce inflammation and peritoneal lesions, whose severity was proportional to the length of the fibers [11,12]. Inhalation studies have allowed us to perform long-term experiments (over 1 year) at lower doses [13], thus confirming the specific pro-fibrogenic effects of CNTs already observed after short-term acute and subacute treatments. Researchers subsequently began paying more attention to extra-pulmonary effects of CNT exposure such as systemic vascular alterations, central nervous system effects, reproductive toxicity and immunotoxicity. In particular, pulmonary exposure to MWCNTs might be associated with an increasing risk of developing cardiovascular diseases [14].

In November 2014, the International Agency for Research on Cancer Working Group (IARC WG) evaluated the carcinogenicity of CNTs and classified a type of MWCNT (MWCNT-7, produced by Mitsui Ltd., Japan) as a group 2B agent (possibly carcinogenic to humans), with limited evidence for other similar MWCNT types and no evidence of carcinogenicity for SWCNT. MWCNT-7 caused peritoneal mesotheliomas in rats and in *p53*^+/−^ mice using injection in the peritoneal cavity (i.p.) [15]. Inhalation of MWNT-7 promoted bronchiolo-alveolar adenoma and carcinoma formation in mice. In an i.p. administration study, MWCNTs similar to MWCNT-7 (1–19 μm in length; diameter 40–170 nm) caused mesotheliomas. The lack of classification of all CNTs as carcinogenic (due to a lack of data) by the IARC WG, with the exception of MWNT-7, was criticized, because it was highlighted that conclusions about MWCNT-7 cannot be extrapolated to all CNTs [16].

The IARC WG reported that some MWCNT can promote the neoplastic growth and tumour progression of lung cells in B6C3F1 mice with a MWCNT lung burden of about 31.2 μg/mouse, a dose which seems relevant for occupational setting (Grosse et al., 2014). Moreover, it was also emphasized that CNTs induce DNA oxidative damage, DNA strand breaks and micronuclei formation. Both SWCNTs and MWCNTs may perturb cell mitosis in human lung epithelial cells. All the above-mentioned mechanisms are certainly relevant to humans [15,17,18].

Both MWCNT-N and MWCNT-7 induced lung tumors and mesotheliomas in rats after a new method of dosing, i.e., trans-tracheal intrapulmonary spraying (TIPS) [19,20]. Recently, Saleh et al. [21] carried out a two-year study using two types of MWCNTs; one thick, straight-type and one thin, tangled-type. Cumulative doses of 0.5 or 1.0 mg tangled MWCNT resulted in lung cancer, whereas the straight-type MWCNT did not exhibit a similar carcinogenic potential.

It should be pointed out that an important aspect of in vivo toxicological studies is the comparability with real-life exposure scenarios. Agglomerates or aggregates often have a greater aerodynamic diameter than individual tubes which then settle in the upper airways, where clearance mechanisms can prevent the lung burden. Accordingly, in studies in which CNTs/CNFs were administered by instillation or intra-tracheal aspiration, aggregation not only affected the deposition site, but also the biological response [22]. In this complex scenario, it becomes difficult to draw unequivocal conclusions about the carcinogenicity of CNTs.

## 3. Physico-Chemical Properties and Related Toxicity

As observed in other nanomaterials, the physico-chemical properties of CNTs influences their toxicity. There are indications that the purity, diameter, length, surface charge, functionalization, and state of aggregation can all determine CNT related toxicity [23,24]. Nonetheless, for a long time, only the dimensional aspect and the “fiber-like” behavior have been considered, rather than the whole pattern of physico-chemical characteristics relevant for their biological effects. Research carried out by our group has shown that the in vitro exposure to MWCNTs causes a progressive decrease of the transepithelial electrical resistance (TEER) of Calu-3 monolayers, thus demonstrating an impairment of epithelial barrier functionality. More specifically, only MWCNTs with fibre-like properties such as long length, reduced solubility and a tendency to aggregate were capable of causing a decrease in TEER [25]. This initial study supports the hypothesis that the toxicity of certain CNT species presents interesting similarities with that of asbestos. Another set of experiments performed in order to define contact-mediated toxicity of MWCNT towards respiratory epithelium cells showed that the addition of MWCNTs to the apical side of Calu-3 human airway epithelial cell monolayers led to the adhesion of some cells to the agglomerates of nanotubes which, after a few days, were covered with epithelial cells. While most of the monolayer was fully viable, several cells adhering to the agglomerates showed clear signs of cytotoxicity—as evidenced by propidium iodide positivity—and positivity to caspase activity, indicating that contact with MWCNTs triggered an apoptotic process [26]. Confocal microscopy revealed that the barrier alteration is partly due to focal cytotoxicity, primarily involving those cells which are in direct contact with CNTs aggregates, and in part to the expansion of the monolayer which would tend to include the aggregates. This data correlates with in vivo observations and can constitute the basis for the definition of the pathogenetic mechanisms triggered by exposure to fibrous nanomaterials with reduced acute toxicity and high biopersistence. Whatever the interpretation of the observed effects, the experiments carried out so far open up other research hypotheses and perspectives. Junctional complexes play a fundamental physiological role in the organism’s defense against pathogens and environmental factors. Interference with this function by fibrous materials, such as CNTs, could represent the first step in establishing a fibrogenic tissue reaction. Vietti et al. [27] have elucidated the sequence of molecular and cellular events which lead to lung fibrosis. According to an Adverse Outcome Pathway (AOP) model, CNTs can directly or indirectly interact with lung fibroblast, with the length and thickness being the main determinant in fibroblast, macrophages and epithelial cell activation [27].

A length of >4–10 μm can almost be considered as a threshold for CNT pathogenicity, and ability to trigger inflammatory responses [11,12,23,24,28,29]. The greater harmfulness of long and straight CNTs (>10 μm) appears to be related to their mechanical interaction with cells, more specifically due to ineffective macrophage phagocytosis of long fibers. Amongst the possible structural determinants of MWCNT-induced toxicity on airway cells, both shape and length seems the most relevant at realistic doses [30].

CNT diameter is another important parameter to consider. MWCNT similar in length (<5 μm) and surface reactivity, but of different diameter (9.4 and 70 nm, respectively) exhibited differential toxicity both in vivo (rat lungs) and in vitro (murine alveolar macrophages), being the thinner MWCNTs more toxic than the thicker ones [31].

Conversely, structural defects on CNTs surface have been associated with inflammation, but not with carcinogenicity [32].

However, the ‘asbestos-like’ pathogenicity of long CNTs can be mitigated through chemical functionalization, e.g., by shortening CNTs through debundling or untangling. Indeed, the degree and type of functionalization appears crucial in the modulation of biological responses and clearance from the organism. The relationship between the degree of functionalization and the elicitation of an inflammatory response coupled with macrophage activation has been recently studied [33,34]. The proportion of tubes with greater nominal lengths and diameters in the mixture seems to affect the severity of genotoxic damage in human epithelial cells caused by the CNTs/CNFs used or produced in U.S. [35].

Biopersistence has been implicated in biological effects. However, SWCNTs can undergo enzymatic degradation in human neutrophils [36]. The authors have suggested that CNTs can be attacked and degraded by endogenous oxidants or oxidizing enzymes, e.g., myeloperoxidase (MPO), found in phagolysosomal fluid [37] and this has been confirmed in MPO-deficient animals, which suffered from greater inflammatory effects and pulmonary fibrosis [38].

A single-walled structure and some types of functionalization favor biodegradation. Carboxylated CNTs, for example, are partially degradable compared to non-functionalized CNTs, which are more resistant to degradation and therefore more biopersistant. However, it is not yet fully clear whether in vivo degradation can prevent the harmful effects of CNTs. Finally, the presence of catalytic residues, such as metallic elements (e.g., Ni, Co, Cu), dramatically increase CNT toxicity [29,39].

## 4. Early Effects in Workers Occupationally Exposed to CNTs/CNF

In spite of the experimental evidence suggesting that exposure to CNTs/CNFs may cause adverse health effects, epidemiological studies involving groups of workers are still exiguous and have many gaps in exposure assessment. This is probably due to the relatively recent consideration of possible health effects in exposure scenarios involving nanotechnology workers, methodological issues and standardization in airborne aerosols measurement, but also by the lack of consistent exposures leading to measurable effects [40]. Table 1 summarizes the main characteristics and findings from available studies in workers occupationally exposed to CNTs/CNFs.

Lee et al. [41] assessed health effects in workers manufacturing MWCNTs along with personal and area monitoring. They found significantly higher concentrations of biomarkers of oxidative stress, such as malondialdehyde (MDA), 4-hydroxy-2-hexenal (4-HHE), and n-hexanal levels in the exhaled breath condensate (EBC) of MWCNT manufacturing workers than among office workers.

In a cohort of workers occupationally exposed to MWCNTs, Shvedova et al. [42] showed significant variations in gene expression profiles between those exposed to elemental carbon (EC) concentrations of 14.42 + 3.8 μg/m^3^ (inhalable) and 2.83 + 0.6 μg/m^3^ (respirable) in the respiratory area, compared to non-exposed workers. In particular, in a subgroup of eight workers directly exposed to aerosols containing MWCNTs during the previous six months, alterations in the profile of genes involved in biological pathways (e.g., cell cycle control, carcinogenesis, apoptosis, cell proliferation) have been highlighted. In a concomitant study on the same subjects [43] it was shown that the exposure to MWCNT was 3 times higher than the recommended exposure level (REL) of 1 μg/m^3^ proposed by the National Institute for Occupational Safety and Health [44]. Exposure parameters were associated with a panel of inflammatory cytokines, interleukins, tumour necrosis factor-α (TNF-α), and KL-6 (a fibrogenesis biomarker) in induced sputum samples. Furthermore, the trasforming growth factor-β1 (TGF-β1) in serum increased in younger workers (<30 years) only.

Vlaanderen et al. [45], who evaluated a panel of serum immune markers and pneumoproteins found a significant increases in ligand 20 marker, fibroblast growth factor and the soluble IL-1 receptor with increasing exposure to MWCNTs. The same researchers carried out a two-step cross-sectional study on the same workers producing MWCNTs (including lab personnel with different exposure levels), and 42 gender and age-matched unexposed controls [46]. They analyzed the associations between exposure to MWCNTs and cardivascular biomarkers, adjusted for confounding variables. A trend in the concentration of biomarkers of endothelial activation (ICAM-1) with increasing exposure to MWCNTs was observed among operators and it was significant across worker categories and across measured MWCNT concentrations.

The epidemiological study promoted by the National Institute for Occupational Safety and Health (NIOSH), which included a cohort of 108 workers from 12 companies in the USA evaluated the association between occupational exposure to CNT/CNF and indicators of early effect in sputum and blood. This study found a positive association between exposure (mean value of inhalable EC background-corrected: 0.24 μg/m^3^) and respiratory allergy and an association of systolic blood pressure with fine particulate matter and heart rate with EC [47]. Beard et al. [48] subsequently examined a panel of sputum and blood biomarkers of inflammation, antioxidant activity, structural remodeling of tissue, pulmonary fibrosis and indicators of endothelial activation. Inhalable CNT/CNF concentration was associated with biomarkers of inflammation, endothelial activation and fibrosis. After adjusting for potential confounders, inhalable EC was associated with biomarkers in sputum, whereas the total inhalable CNT/CNF structures were associated with blood biomarkers. Schubauer-Berigan et al. [49] re-assessed 102 workers of the above cohort. Each participant provided a blood sample which was incubated with and without two bacterial endotoxins. The stimulant:null response ratio for each biomarker was analyzed using multiple linear regression, adjusted for confounders along with a set of 46 biomarkers. Subjects belonging to the lowest exposure tertile (as CNT/CNF counts) showed a tendency towards higher biomarker ratio levels as compared to higher exposed subjects. However, some inverse association between CNT/CNF counts and stimulant:null ratios of several individual biomarkers were also seen.

**Table 1 nanomaterials-11-00745-t001:** Studies on workers occupationally exposed to carbon nanotubes (CNT)/carbon nanofibers (CNF) and main findings in biomarker of effects (biochemical and/or functional parameters) among exposed subjects. PBZ: personal breathing zone; R: respirable fraction; I: inhalable fraction; EC: elemental Carbon.

Reference	Elemental Carbon μg/m^3^	No. Exposed vs. Unexposed	Main Findings
Lee et al., 2015 [41]	(PBZ sampling) CNT 6.2–9.3	9/4	Increased 4-hydroxy-2-hexenal (4-HHE), n-hexanal and malondialdehyde (MDA) in exhaled breath condensate
Shvedova et al., 2016 [42]	MWCNT 2.8 R 14.4 I	8/7	Alterations in ncRNA and mRNA expression profiles, involving target genes with roles in cell cycle regulation/progression/control, apoptosis, cell proliferation and carcinogenetic pathways
Fatkhutdinova et al., 2015 [4]; 2016 [43]	MWCNT 0.7–2.8 R	10/12	Increased concentration of IL-1β, IL-4, IL-5, IL-6, IL-8 and TNF-α in sputum; increase of KL-6 in sputum; increase of IL-1β, IL-4, IL-10 and TNF-α in serum and TGF-β1 (only in workers <30 year old)
Vlaanderen et al., 2017 [45]	MWCNT 45–57 I	22/39	Increase in C-C motif ligand 20, soluble IL-1 receptor II
Kuijpers et al., 2018 [46]	MWCNT 1–7 (Lab)/45–57 (Op) I	22/42	Increase in ICAM-1 across worker categories and across measured GM MWCNT concentrations
Schubauer-Berigan et al., 2018 [47]	Median: 0.24 I EC 0.096 R EC (mean: 6.22 I EC; 1.00 R EC)	108	Positive association with respiratory allergy with inhalable elemental carbon (EC) airborne concentration and number of years worked; association of systolic blood pressure with fine particulate matter and heart rate at rest with EC
Beard et al., 2018 [48]	Median: 0.24 I EC 0.096 R EC (mean: 6.22 I EC; 1.00 R EC)	108	Association of CNT/CNF metrics with type IV collagenase/matrix metalloproteinase-2 (MMP-2), IL-18, glutathione peroxidase (GPx), myeloperoxidase, superoxide dismutase (SOD) in sputum; association with MMP-2, matrix metalloproteinase-9, metalloproteinase inhibitor 1/tissue inhibitor of metalloproteinases1, GPx, SOD, 8-hydroxy-2′-deoxyguanosine, endothelin-1, fibrinogen, ICAM-1, vascular cell adhesion protein 1 and von Willebrand factor in blood.
Schubauer-Berigan et al., 2020 [49]	Structure/cm^3^ (TEM): 0.219 (mean); 0.0087 (median)	102	Significant differences (*p* < 0.05) for Haptoglobin and a pattern of citokines and growth factors (e.g., MMP9, SCF, TIMP1, VEGF) in plasma

## 5. Exposure Assessment and Measurement Methods

Occupational exposure to particulates containing CNT/CNF is potentially possible throughout the whole life cycle of materials, like during engineering, production of raw material, composite materials production, and secondary manufacture and use [40,50].

For instance, during the engineering phase, the target population is represented by academics and workers involved in research and development; in such a scenario, emission is mainly accidental, leading to potentially high concentration and can be reliably described by the usual exposure metrics (e.g., in mass concentration or number concentration of CNT/CNF). Other stages of life-cycle are characterized by a lower exposure potential (e.g., in the primary manufacture). However, it is worth mentioning that the low rate pattern of exposure characterizing the final steps (i.e., transformation, composite production, maintenance) makes the risk associated with such an exposure scenario potentially negligible [40].

CNT sampling can be performed by using cassettes with quartz filters and CNT analysis can be carried out by thermo-optical analysis with a flame ionization detector (FID) or with mixed cellulose esters (MCE) filters for analysis by transmission electron microscopy (TEM) to ensure specificity. Existing measurements have been made over short periods and on a limited number of operations/exposure situations.

The results of such measurement strategy are uncertain, since the method does not specifically measure CNTs, but also other airborne particulate matter. In order for the measurements to be valid, it is necessary to distinguish CNTs from the environmental background.

In field studies, different types of sampling equipment have been used, making the findings difficult to compare. A more exact metric is represented by the specific chemical mass concentration of total Carbon and elementary Carbon, as specified by the NIOSH 5040 method to establish a recommended exposure limit (REL) [51,52,53]. Even though assessing structures containing CNT/CNFs with electron microscopy represents a sufficiently selective and sensitive method to quantify exposure, it is still premature to propose mean values and ranges, due to important variations in counting protocols and heterogeneity of sampled material.

Exposure scenario involving workers handling composites (CF impregnated with epoxy resins) to build aircraft components, revealed that airborne fibers concentrations were significantly higher in personal samplings (median value 7.01 ff/L) than in stationary samplings (median value 1.93 ff/L), with particles and fibers >20 µm. Interestingly, characterization of airborne fibers has been accomplished by using SEM coupled with X-ray microanalysis system (SEM-EDXA) [54].

The release of fibrous structures can also occur during experimental processes in R&D facilities, because of accidental events occurring during production and use processes. In closed production lines, emission and exposure can only occur after the reactor is opened for the collection of test samples or in case of leaks or spills. The fragmentation and abrasion of nanotechnological products and other activities, such as recycling and waste treatment, represent other potential exposure scenarios.

It has been pointed out that CNT measurement strategies require harmonization. Almost 50% of the studies assessed by Guseva Canu et al. [40,50] reported the results of measurement of EC mass concentration using different methods and aerosol fractions, whereas 85% of field studies found values exceeding the NIOSH Recommended Exposure Level (REL) of 1 μg/m^3^.

Most of the exposure measurements in different workplaces in which CNTs are produced or handled rely on the traditional measurements of powders based on mass concentration metrics. The measured levels are in the order of tens of μg/m^3^, but higher concentrations have been reported in some activity (Table 2, modified from [40]).

These field studies revealed that in the atmosphere of working environments CNT/CNF are usually found as aggregates of (sub)-micrometric dimensions and rarely as single fibers [55,56,57,58]. The quantification of CNT agglomerates and/or CNT containing fibers is becoming increasingly common. The greatest mean CNT mass concentrations was found during non-enclosed activities, including packaging, cleaning, sieving, during extrusion and pelletizing. In general, a ranking of exposure potential can be estimated, with technicians experiencing the highest level of exposure, followed by engineers, then by chemists.

Dahm et al. [59] studied, at the company-level and at the single worker level, the determinants that increased exposure. These authors demonstrate that inadequate operational and engineered controls can affect at company level the release of CNT/CNF as well as their diameter and length, combined with quantities of material handled every day. At worker level, it was found that handling dry-powdered form of CNT/CNF in quantities of >1 kg, job titles (e.g., engineer vs. other tasks, e.g., handling powders), working in a ventilated or unventilated environment, all represented consistent determinants. By regression models, the authors obtained insights into the best exposure predictors to CNT/CNF for each exposure metric, but finally suggested the integration of their model with other methods.

## 6. Hygienic Standards at the Workplace

In order to carry out the risk assessment at workplace and in the environment, systematic data on acceptable levels of CNT exposure is required. Table 3 summarizes the hygienic standard for CNTs and CNFs.

However, the development of hygienic standards has encountered various difficulties due to the variety of types of CNTs, the complexity of their identification and quantification in the work environment and preliminary findings on biological effects. The British Standards Institution (BSI) first suggested a maximum permitted level of 0.01 fibers/m^3^ relying on the similarities with asbestos [60]. It is worth mentioning that all the above limits aim to protect the exposed population from the risk of chronic respiratory outcomes, such as lung fibrosis, but not from carcinogenetic events.

To determine safe exposure levels, we can rely on the extrapolation of the results gathered in animal studies to estimate an observed adverse effect level (OAEL) and further establish factors of uncertainty.

Since 2010, the NIOSH have proposed a REL of 7 μg/m^3^ as 8-h TWA based on the EC determination representing the upper limit of quantitation for NIOSH Method 5040 [61]. However, NIOSH recognized that workers may still have an excess risk of developing early-stage lung fibrosis if exposed over a full working lifetime to the proposed REL. The NIOSH starting point for the quantitative risk assessment was data derived from animal studies investigating non-malignant pulmonary outcomes. Many animal studies have observed lung fibrosis (characterized by persistence or even progression after exposure cessation) that developed early (i.e., 28 days after exposure) in response to relatively low-mass lung doses [8,62,63,64,65,66,67]. These data represent a consistent basis for a REL, whereas findings regarding carcinogenic and cardiovascular effects were not considered in the quantitative risk assessment by inhalation exposure. NIOSH considered lung inflammation and fibrosis found in short-term and subchronic animal studies as the endpoints relevant to humans, i.e., health effects that have been (or could be) observed following occupational exposure to inhaled particles and fibers. Uncertainties remain about the extent of functional deficits and their health significance among workers.

Working lifetime exposure concentrations were calculated based on estimates of either the deposited/retained alveolar lung dose of CNT assuming an 8-h TWA exposure during a 40-h workweek, 50 weeks per year, for 45 years, or on BMD modelling of the subchronic animal inhalation studies with MWCNTs [63,64]. It was estimated that a working lifetime exposure of 0.2–2 μg/m^3^ (8-h TWA concentration) is associated with a 10% excess risk of early-stage adverse lung effects of minimal severity (grade 1) whereas 0.7–19 μg/m^3^ could cause mild grade lesions. A LOAEL between 4–18 μg/m^3^ and a NOAEL between 1–4 μg/m^3^ for an 8-h TWA was thus calculated.

After reviewing the animal and other available toxicological data concerning the potential adverse respiratory effects of CNT/CNF, NIOSH provided a quantitative risk assessment based on dose-response data gathered in animal studies [66,68]. Considering improvements in sampling and analytical techniques, NIOSH recommended 1 μg/m^3^ EC as an 8-hr TWA REL for respirable mass concentration, to reduce the risk for pulmonary inflammation and fibrosis [44]. Thus, although data from animal studies dealing with CNFs are more limited as compared to CNTs, physico-chemical similarities, and findings of acute pulmonary inflammation and interstitial fibrosis, suggest maintaining exposure to CNFs at a REL of 1 μg/m^3^ EC [44].

The risk assessment carried out by the Belgian company Nanocyl, based on the lowest concentrations of adverse effects observed in subchronic inhalation experiments (90 days) on rats, gave a weighted average concentration of 2.5 mg/m^3^ as 8-h TWA for its MWCNTs. In this study, Nanocyl NC 7000—equivalent to the EU reference material MWCNT NM-400—did not cause systemic toxicity [63]. However, at lower doses (0.5–2.5 mg/m^3^) this MWCNT induced multifocal granulomas, diffuse histiocytic/neutrophilic inflammation and lipoproteinosis in the lungs and regional lymph nodes. As a result, a working lifetime equivalent concentration of 0.0011 mg/m^3^ was extrapolated by the corresponding Lowest Adverse Effects Concentration (LOAEC) found in this study. The consistency of such threshold was compared to the results of MWCNT measurements campaigns at workplace [57], and different release scenarios of composite materials using MWCNT as fillers [69,70,71]. In spite of a low tendency to form dusts, the need to comply with industrial hygiene practices during handling and processing was nonetheless recognized [63,71].

The Bayer Company has proposed a time weighted average occupational exposure limit (OEL) of 50 μg/m^3^ for Baytubes™, a more flexible MWCNT type with the tendency to form tube agglomerates [64].

Finally, the Japanese National Institute of Industrial Science and Technology derived an occupational exposure limit (OEL) for all CNT of 30 μg/m^3^ [72]. Interestingly, Japanese studies indicated that CNT does not cause direct genotoxic damage, but rather a secondary genotoxicity resulting from free radicals’ generation following inflammatory processes. Interestingly, such an effect is characterized by a mechanistic threshold, which has relevant implications regarding the set-up safety limits.

## 7. Conclusions

CNTs are currently protagonists of a debate regarding their long-term toxicity and carcinogenicity. Several physicochemical parameters, including shape, state of agglomeration/aggregation, surface properties, impurities and density, may influence toxicity. However, some conclusions have been achieved at exposure concentrations that are unlikely to occur in workplace settings [22].

Animal studies showed that some rigid MWCNTs of a certain length and aspect-ratio are able to induce mesothelioma by intraperitoneal injection [11,73], whilst tangled MWCNT did not. Despite data reported on existing reviews, a consensus regarding CNT toxicity has yet to be attained. Existing studies report conflicting results, which can be partly attributed to the lack of standardized protocols and to the variability of CNT types. This indicates that the biological responses to CNT exposure are modulated by physico-chemical properties and that CNTs cannot be considered as a single, unified material entity but instead should be understood as a class of materials with varying properties that may elicit distinct biological outcomes both in vitro and in vivo. Discrepancies regarding the definition of the toxicological profile of CNTs due to the lack of standardized assessment methodologies and the variability of the properties of the samples used in toxicological tests make risk assessment difficult. The hazard of short (<4–5 μm), monodispersed and functionalized CNTs is certainly different from that of CNTs used for industrial applications [74].

Because this is the case, a possible risk management strategy could be to choose alternative MWCNTs lacking critical properties which trigger toxicity, an approach currently described as “safety by design” [75,76].

Recently, the International Chemical Secretariat added CNTs to the SIN list of chemicals that should be restricted or even banned in the EU [2]. However, an international group of experts raised issues on the consequences on the innovation development of this “scientifically unjustified proposal” [3,76]. The authors also claimed that the proposal of re-evaluation asked by the IARC Advisory Group is still pending [77,78].

It should be recognized that the risk assessment for CNTs/CNFs is particularly challenging, for the following reasons: (1) there is no consensus on which metric is more appropriate for health effects; (2) CNTs may be present in a large number of variants that may have different degrees of toxicity; (3) toxicological data is often insufficient, but indicates a possible risk of genotoxicity, inflammation and pulmonary fibrosis following CNT inhalation at relatively low doses; (4) Exposure measurements are limited to the initial stages of the life cycle and cover a limited set of tasks.

Field studies have focused on the release of CNTs/CNFs in the workplace atmosphere rather than on individual exposure [50,79]. The lack of harmonization in measurement strategies, analysis, and reporting of measurement data hinders exposure and risk assessments. Mass-based approaches allow us to assess the exposure potential and emission sources, but other parameters, like the airborne concentration of fibers or, better, CNTs/CNFs structures quantified by electron microscopy can help in exposure characterization.

A probabilistic approach has been recently suggested, in order to estimate the likelihood of risk for a selected occupational scenario and propose a realistic benchmark concentration (expressed as mass concentrations) for occupational exposure to CNTs [80]. Although pragmatic and useful, such an approach reveals its limitations when facing hazard heterogeneity and in predicting long-term health effects.

Cross-sectional investigations have suggested an association between subtle human health effects and CNTs/CNFs workplace exposure. However, due to the long latency required for overt health effects, cohort studies with a sufficient follow-up period are needed. This requires that workplace determinants that contribute to exposure should be clearly identified, and consistent methods of measurements should be proactively investigated and validated [81].

In this complex and uncertain scenario, all types of CNTs and CNFs should be considered as respiratory hazards, and occupational exposure should be kept as low as possible. Current knowledge on health effects or exposure levels during the handling of CNTs is increasing and qualitatively improving. Until a better risk characterization has been established, a precautionary principle should be adopted in the production and use of CNTs/CNFs and in the processing of materials containing CNTs/CNFs [3].

## Figures and Tables

**Table 2 nanomaterials-11-00745-t002:** Examples of results of field measurements to CNT/CNF in some occupational environments and related metrics (From: [40], modified).

Material	Activity	Sample	Exposure Range	Operations Generating Emissions
**Total Carbon (µg/m^3^)**				
CNF	R&D laboratory	area	15–1094	wet saw cutting of CNF composite
		PBZ	64–1094	wet saw cutting of CNF composite
	Industry	area	31–1839	drying CNFs
**Elemental Carbon (µg/m^3^)**				
CNF	R&D laboratory	area	ND–1900	wet saw cutting inside ventilated booth
		PBZ	ND–1000	wet saw cutting with no control
	Industry	area	ND–476	transfer to mixing vessel in ventilated, negative pressure room
		PBZ	ND–7.54	weighing, mixing and sonication
		area (respirable)	11.3–13	synthesis in CVD reactor
		PBZ (respirable)	27.3–80	synthesis in CVD reactor
MWCNT	R&D laboratory	area	*NM*	NA
		PBZ	*NM*	NA
	Industry (production)	area	ND–470	cleaning of deposits with no control
		PBZ	ND–7.4	all arc discharge-MWCNT production operations
	Industry (production)	area	ND–1.89	batch mixer use with chemical hood and PPE
		PBZ	ND–7.86	batch mixer use with chemical hood and PPE
SWCNT	Industry	area	ND–39	loading flasks with CNTs
		PBZ	ND–38	harvesting CNTs from reactor
		area (respirable)	Media = 0.26	all SWCNT production operations
		PBZ (respirable)	media= 0.05	all SWCNT production operations
**CNT/NCF (structures/cm^3^)**				
CNF	R&D laboratory	PBZ	*NM*	*NA*
	Industry	area	0.003–0.295	transferring CNF
		PBZ	0.07–1.16	transferring CNF
MWCNT	R&D laboratory	area	ND–172.9	blending CNTs with no control
		PBZ	ND–193.6	blending CNTs with no control
	Industry	area	ND–11	sieving, pouring, weighing CNTs
		PBZ	ND–2	all arc discharge-MWCNT production operations
SWCNT	Industry	area	0.007–0.012	CNT synthesis, harvesting, reactor clean-out
		PBZ	0.003–0.01	CNT synthesis, harvesting, reactor clean-out

CNF = carbon nanofibers; CNT = carbon nanotubes; SWCNT = Single-walled carbon nanotube; ND = not detected (below analytical limit of detection); NM = not measured; NA = not available; LOQ = limit of quantification; PBZ = personal breathing zone sample; resp = respirable fraction; CVD = chemical vapor deposition process.

**Table 3 nanomaterials-11-00745-t003:** Hygienic standard at workplace for CNTs and CNFs by inhalation route.

Institution	Concentration (Number/Mass)	Interpretation	Year
British Standards Institution (WEL)	0.01 fibres/mL	Fibrous nanomaterials with highaspect ratios (>3:1) and length >5000 nm (>5 μm)	2007
Nanocyl	0.0011 mg/m^3^	8-h TWA (working lifetime equivalent)	2009
Bayer (OEL)	50 μg/m^3^	8-h TWA (Baytubes™)	2010
US NIOSH (REL)	7.0 μg/m^3^	8-h TWA	2010
Dutch Social and Economic Council (OEL)	0.01 fibres/cm^3^ (10.000 f/m^3^) fibres	SWCNT or MWCNT for which asbestos like effects are not excluded	2012
US NIOSH (REL)	1.0 μg/m^3^ (r EC)	8-h TWA	2013
US OSHA (recomm.)	1.0 μg/m^3^ (r EC)	8-h TWA	2013
Nakanishi (OEL)	30.0 (SW)–80 (MW) μg/m^3^	8-h TWA	2015
Swiss Accident Insurance Funds	0.01 fibres/mL	8-h TWA	2018

WEL: workplace exposure limit; TWA: Time weighted Average (concentration); OEL: occupational exposure limit; REL: Recommended Exposure Limit.

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
