# Peer review of "Occupational Exposure to Carbon Nanotubes and Carbon Nanofibres: More Than a Cobweb"

_nanomaterials, 2021, doi:10.3390/nano11030745_

Round 1

Reviewer 1 Report

This is a very interesting Review work on the properties and health hazards of nanomaterials. Very well organized and presented. The final Conclusions are loud and clear proposing possble regulatory measures to protect the environment and  workers from health issues.

Author Response

Dear Reviewer,

Thank you very much for considering our manuscript as suitable for publication without changes.

Best regards.

Prof. Enrico Bergamaschi

Reviewer 2 Report

Dear authors:

The present manuscript studies the hazards associated with carbon nanotubes and carbon nanofibres. Not as single materials but as a family with varying properties. I find the study interesting, well-conducted, and necessary in the field of nanomaterials such as nanotubes and nanofibres. I, therefore, consider it to be of particular relevance to the field of nanomaterials and the dangers associated with the scale and reactivity of nanomaterials. The paper is well developed and well described and gives a general idea of the risks and hazards. I, therefore, consider it suitable for publication in its present form

Author Response

Dear Reviewer,

Thank you very much for considering our manuscript of relevance for the nanomaterials field and suitable for publication without changes.

Best regards.

Prof. Enrico Bergamaschi

Reviewer 3 Report

In this review the authors make a critical discussion on different aspects related to the long-term toxicity and carcinogenicity of CNTs and CNFs They present conclusions of different studies regarding the influence of different parameters (length, shape, surface, aggregation etc.) of CNT/CNF on their possible hazardous effects in humans, considering some correlations between occupational inhalation exposure during different stages of CNT/CNF R&D and various risks on human health. Sampling and analysis methods used for exposure measurements were also briefly discussed.

I have few minor observations:

Page 5: Write "CALU-3" or "Calu-3".

All abbreviations should be explained when they appeared for the first time in the text. e.g. NIOSH (Page 9), LOAEC  (Page 14), TWA and OEL as food note of table 3 (page 13) and not at page 14, WEL as table 3 food note, etc.

Page 12: "The Authors" should be written "The authors.."

Author Response

Dear Reviewer,

Thank you very much for considering our manuscript suitable for publication with minor changes.

In the attached .pdf file you can see the explanation of abbreviations requested.

Best regards.

Prof. Enrico Bergamaschi
